# Efficacy and safety of diazoxide for treating hyperinsulinemic hypoglycemia: A systematic review and meta-analysis

**Xiaohong Chen** ⬤ *[⊚], **Lifang Feng**[⊚], **Hui Yao, Luhong Yang, Yuan Qin**

Department of Endocrinology and Metabolism, Wuhan Children's Hospital, Tongji Medical College, Huazhong University of Science and Technology, Wuhan, China

⊚ These authors contributed equally to this work.
* cxhdaifu@163.com

**Data Availability Statement:** All relevant data are within the manuscript and its Supporting Information files.

## Abstract

Diazoxide is the first-line drug for treating hyperinsulinism and the only pharmacological agent approved for hyperinsulinism by the Federal Drug Administration. This systemic review and meta-analysis aimed to investigate the efficacy and safety of diazoxide for treating hyperinsulinemic hypoglycemia (HH). The meta-analysis of the efficacy and safety of diazoxide in treating HH was performed by searching relevant studies in the PubMed, Embase, and Cochrane databases. The findings were summarized, and the pooled effect size and its 95% confidence interval (CI) were calculated. A total of 6 cohort studies, involving 1142 participants, met the inclusion criteria. Among the cohort studies, the pooled estimate of the response rate of diazoxide therapy was 71% (95% CI 50%–93%, $P_{heterogeneity}$< 0.001, $I^2$ = 98.3%, $P_{effect}$< 0.001). The common side effects were hypertrichosis (45%), fluid retention (20%), gastrointestinal reaction (13%), edema (11%), and neutropenia (9%). Other adverse events included pulmonary hypertension (2%) and thrombocytopenia (2%). This meta-analysis suggested that diazoxide was potentially useful in HH management; however, it had some side effects, which needed careful monitoring. Furthermore, well-designed large-scale studies, such as randomized controlled trials, might be necessary in the future to obtain more evidence.

## Introduction

Hyperinsulinemic hypoglycemia (HH) describes the condition and effects of low blood glucose caused by excessive insulin. Many cases are reported in childhood as a congenital disorder [1]. Congenital hyperinsulinemia (CHI) is the most common and serious cause of persistent hypoglycemia in newborns and children. It occurs between 1:2500 and 50,000 live births [2]. CHI is characterized by excessive secretion of insulin by pancreatic B cells, which is the most common cause of persistent hypoglycemia in infancy [2]. CHI is a heterogeneous disease in clinical manifestations, imaging, histology, and genetics. So far, mutations in more than 10 different genes (*ABCC8*, *KCNJ11*, *GLUD1*, *GCK*, *HADH*, HK1, CACNA1D, FOXA2, UCP2, *SLC16A1*, *HNF4A*, *HNF1A*, *PMM2*, and *PGM1*) have been reported in the genetic etiology of CHI [3–6]. CHI has three main histological types: focal, diffuse, and atypical [7,8].

**Funding:** The authors received no specific funding for this work.

**Competing interests:** The authors declare that they have no conflict of interest.

Diazoxide is the first choice for treating CHI. It is a nondiuretic benzothiadiazine originally formulated as a peripheral vasodilator to reduce severe hypertension by smooth muscle relaxation [9]. It was first used to treat CHI in the 1960s [10] and has been the primary treatment since then. Diazoxide is the only drug approved for this indication in the United States, Canada, the United Kingdom, the European Union, China, Australasia, and Japan, [11]. Diazoxide acts by binding to the sulfonylurea receptor-1 subunit in the ATP-sensitive K+ ($K_{ATP}$) channel, causing the channel to open and increase its permeability to potassium ions. This results in the excessive polarization of beta cells, followed by the inhibition of $Ca^{2+}$-dependent insulin secretion [12].

Several clinical studies investigated the efficacy and safety of diazoxide for treating CHI [13–18]. However, the results remain ambiguous. Meissner et al. [13] reported that 47/114 (41.2%) patients responded to diazoxide. Hu et al. [14] found that 36/44 (81.8%) patients were responsive to diazoxide treatment. This meta-analysis based on six cohort studies was conducted and aimed to investigate the efficacy and safety of diazoxide for treating CHI.

## Materials and methods

The present meta-analysis was conducted according to the Preferred Reporting Items for Systematic Reviews and Meta-analysis guidelines [19].

### Search strategy

The PubMed, Embase, and Cochrane databases were searched for studies published up to January 2021, using the following terms: "hyperinsulinemic hypoglycemia," "Congenital hyperinsulinism," "hypoglycemia," "neonates," "infants," "children," and "diazoxide" with all possible combinations. The search strategy for the Embase database was included in the supplementary document in S1 Table. Using these parameters, all eligible studies were filtered out, and their reference lists were viewed for more available studies.

### Study selection

The inclusion criteria were as follows: (1) cohort studies focusing on the efficacy and safety of diazoxide in treating HH; (2) studies reporting clinical outcomes such as response rates and complications; (3) studies available with the full text; and (4) if more than one study was published using the same case series, selection of the study with the largest sample size. The exclusion criteria were as follows: (1) experimental studies of animal models or cell lines; (2) similar studies involving repetition of patients; and (3) abstract or inappropriate types of publications, such as reviews, guides, or case reports. The selection had no language restrictions. Two investigators independently assessed the eligibility of each study and resolved any differences through discussion.

### Data extraction and quality assessment

Two investigators (Xiaohong Chen and Lifang Feng) independently extracted all available data from the included studies based on the description provided by the authors of these studies. Subsequently, any differences were resolved through discussions with the third author. The following information was extracted from all relevant studies: first author, year of publication, country, sex, mean age, number of patients, follow-up time, and assessment results. The quality of the cohort study was assessed using the 9-star Newcastle–Ottawa Scale [20]. High-quality studies were defined as a study with more than 7 stars [20].

## Statistical analysis

The meta-analysis was conducted using Stata 12 (Stata-Corp, TX, USA). A meta-analysis of cohort studies based on the random-effects model was conducted to evaluate the clinical response rate (patients responsive to diazoxide/all patients treated with diazoxide) and complications using effect size (response or incidence rate) and its corresponding 95% confidence interval (CI). The heterogeneity of the studies was assessed using Cochran's Q test and quantified using the $I^2$ statistic (considered high heterogeneity for $I^2 > 50\%$) [21,22]. Publication bias was evaluated by the visual inspection of the symmetry of the funnel plot and assessment of Begg's and Egger's tests [23]. The trim-and-fill analysis was applied in the case of any publication bias [24].

# Results

## Study selection

The electronic search identified 348 studies. Three additional studies were found by hand searching from the reference lists of other review studies. According to the inclusion criteria, 227 studies remained after removing the duplicates. Subsequently, 193 irrelevant studies were excluded. Of the remaining 34 studies, 15 were letters, reviews, and meta-analyses, and hence excluded. The remaining 19 studies were systematically reviewed and qualified for full-text reading. After full-text reading, five studies not focusing on children, three lacking usable data, and five case reports were excluded. Finally, 6 studies involving 1142 patients were included in the present meta-analysis [13–18]. The flow chart of the selection of studies and reasons for exclusion is presented in Fig 1.

## Characteristics of the studies

The main characteristics of the eligible studies are shown in Table 1. Six cohort studies were included in the meta-analysis. The studies were performed in 4 countries (Germany, China,

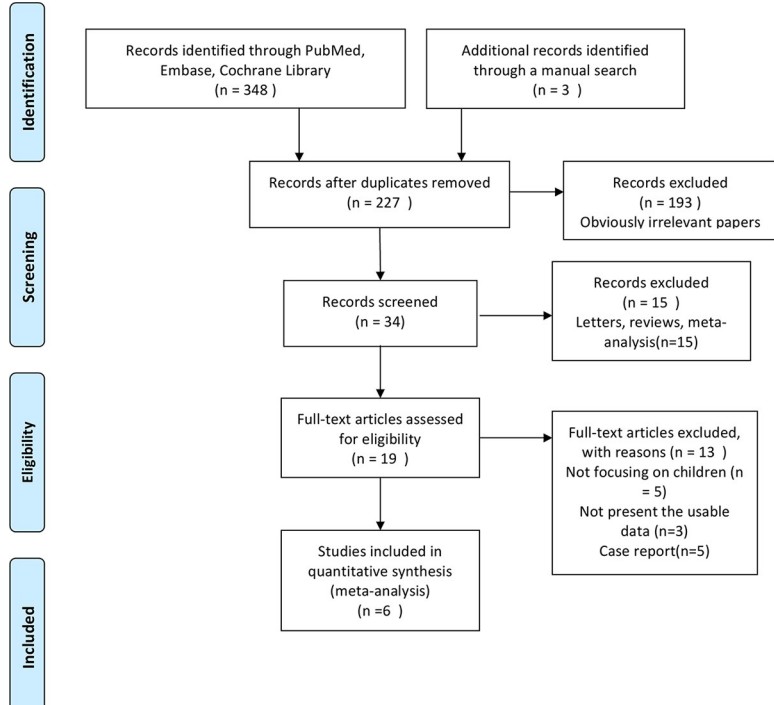

**Fig 1. Flow diagram of study identification.**

**Table 1. Characteristics of the studies included in this meta-analysis.**

| Authors/Year of publication | Country | Male (%) | Age | Birth Weight (g) | Diazoxide (mg/(kg · day)] | Number of patients | Study design | Follow-up | Outcomes assessed |
|---|---|---|---|---|---|---|---|---|---|
| Meissner/2003 [12] | Germany | 52.6 | 1 D −17 Y | 3670 | 15 | 114 | Retrospective cohort | 6.6Y | Respond to diazoxide |
| Hu/2012 [13] | China | 68.2 | 1 D−2 Y | 2200 −5100 | 5−15 | 44 | Retrospective cohort | NA | Respond to diazoxide, Fluid retention, Gastrointestinal reaction, Hypertrichosis |
| Wang/2017 [14] | China | 55.1 | NA | 1900 −5800 | NA | 140 | Retrospective case | NA | Respond to diazoxide |
| Fukutomi/2018 [15] | Japan | 61.2 | 0 M −15 Y | NA | 0.3−17.4 | 384 | Special survey | 7Y | Respond to diazoxide, Pulmonary hypertension, Edema, Thrombocytopenia, Fluid retention, Gastrointestinal reaction, Hypertrichosis |
| Herrera/2018 [16] | USA | 56.3 | 8−161 D | 2350 −3700 | 10−15 | 295 | Retrospective cohort | NA | Neuthropenia, Pulmonary hypertension, Edema, Thrombocytopenia |
| Thornton/2019 [17] | USA | 58.8 | 1 D −17 Y | 580−6600 | 2−12 | 165 | Retrospective cohort | NA | Respond to diazoxide, Neuthropenia, Pulmonary hypertension |

D: Day; Y: Year; NA: Not available.

Japan, and the United States), and the study size ranged from 44 to 384 patients. The patients' age ranged from 1 day to 17 years. The methodological quality of cohort studies included in the meta-analysis is shown in Table 2. The quality of the cohort studies included in the meta-analysis was generally high: two studies had seven stars, three studies had six stars, and one study had five stars.

## Quantitative synthesis

**Response to diazoxide.** The response to diazoxide in patients with congenital hyperinsulinism (CHI) depends on the mutational status. Typically patients with focal CHI or patients with homozygous or compound heterozygous mutations in the K-ATP channel do not respond to diazoxide. Five studies provided outcomes regarding the response to diazoxide in

**Table 2. Methodological quality of observational studies included in the meta-analysis[1].**

| First author | Representativeness of the exposed cohort | Selection of the unexposed cohort | Ascertainment of exposure | Outcome of interest not present at the start of the study | Control for important factor or additional factor | Outcome assessment | Follow-up long enough for outcomes to occur | Adequacy of the follow-up of cohorts | Total quality scores |
|---|---|---|---|---|---|---|---|---|---|
| Meissner/ 2003 [12] | ☆ | — | ☆ | ☆ | — | ☆ | ☆ | ☆ | 6 |
| Hu/2012 [13] | ☆ | — | ☆ | ☆ | — | ☆ | ☆ | ☆ | 6 |
| Wang/2017 [14] | ☆ | — | ☆ | — | — | ☆ | ☆ | ☆ | 5 |
| Fukutomi/ 2018 [15] | ☆ | ☆ | ☆ | ☆ | — | ☆ | ☆ | ☆ | 7 |
| Herrera/ 2018 [16] | ☆ | — | ☆ | ☆ | — | ☆ | ☆ | ☆ | 7 |
| Thornton/ 2019 [17] | ☆ | — | ☆ | ☆ | — | ☆ | ☆ | ☆ | 6 |

[1] A study could be awarded a maximum of one star for each item except for the item Control for important factor or additional factor.

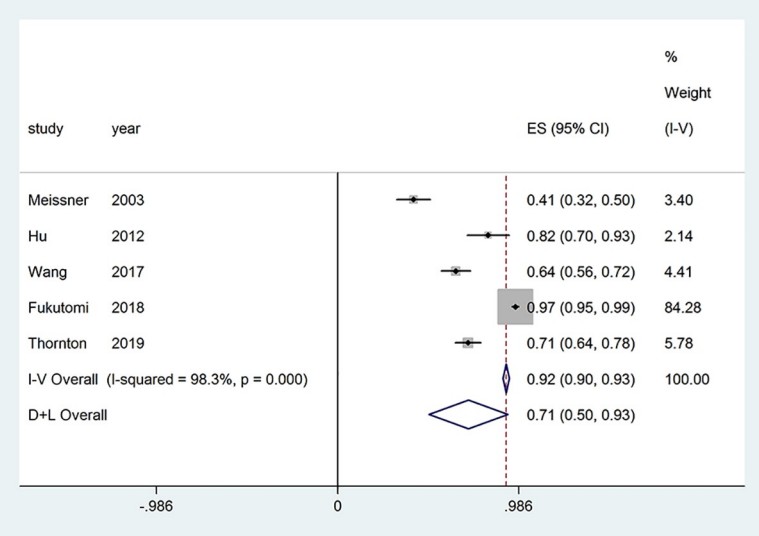

**Fig 2. Effect of diazoxide on patients with hyperinsulinemic hypoglycemia.**

patients with HH and were included in the meta-analysis. Significant evidence of heterogeneity was found among the studies ($P_{\text{heterogeneity}} < 0.001$, $I^2 = 98.3\%$); therefore, a random-effects model of analysis was used. The pooled proportion of patients who were responsive to diazoxide was 71% (95% CI = 50%–93%, $P_{\text{effect}} < 0.001$) (Fig 2).

**Edema.** Two studies provided outcomes regarding edema in patients with HH and were included in the meta-analysis. Significant evidence of heterogeneity among the studies was found ($P_{\text{heterogeneity}} < 0.001$, $I^2 = 95.1\%$, $P_{\text{effect}} < 0.001$); therefore, a random-effects model of analysis was used. The pooled proportion of patients who had edema was 11% (95% CI = 0–22) (Fig 3A).

**Fluid retention.** Two studies provided outcomes regarding fluid retention in patients with HH and were included in the meta-analysis. Significant evidence of heterogeneity among the studies was found ($P_{\text{heterogeneity}} < 0.001$, $I^2 = 96.5\%$, $P_{\text{effect}} = 0.008$); therefore, a random-effects model of analysis was used. The pooled proportion of patients who had fluid retention was 20% (95% CI = –18 to 59) (Fig 3B).

**Gastrointestinal reaction.** Two studies provided outcomes regarding gastrointestinal reaction in patients with HH and were included in the meta-analysis. Significant evidence of heterogeneity among the studies was found ($P_{\text{heterogeneity}} < 0.001$, $I^2 = 93.5\%$, $P_{\text{effect}} = 0.045$); therefore, a random-effects model of analysis was used. The pooled proportion of patients who had gastrointestinal reaction was 13% (95% CI = –13 to 39) (Fig 3C).

**Hypertrichosis.** Two studies provided outcomes regarding hypertrichosis in patients with HH and were included in the meta-analysis. Significant evidence of heterogeneity among the studies was found ($P_{\text{heterogeneity}} < 0.001$, $I^2 = 99.3\%$, $P_{\text{effect}} < 0.001$); therefore, a random-effects model of analysis was used. The pooled proportion of patients who had hypertrichosis was 45% (95% CI = –27 to 117) (Fig 3D).

**Neutropenia.** Two studies provided outcomes regarding neutropenia in patients with HH and were included in the meta-analysis. Significant evidence of heterogeneity among the studies was found ($P_{\text{heterogeneity}} < 0.001$, $I^2 = 92.1\%$, $P_{\text{effect}} = 0.005$); therefore, a random-effects model of analysis was used. The pooled proportion of patients who had neutropenia was 9% (95% CI = 0–19) (Fig 3E).

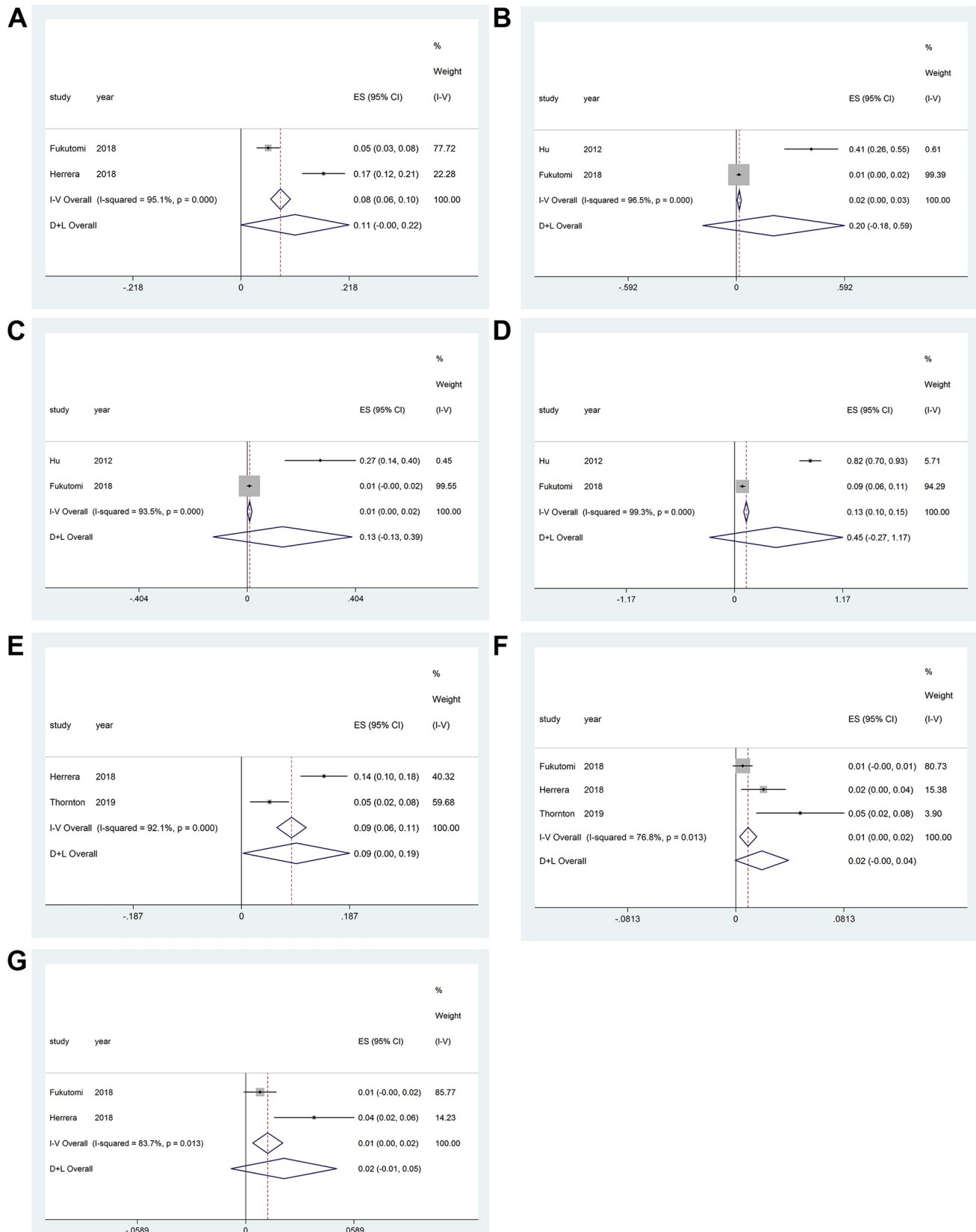

**Fig 3. Safety of diazoxide in patients with hyperinsulinemic hypoglycemia.** (A) Edema; (B) fluid retention; (C) gastrointestinal reaction; (D) hypertrichosis; (E) neutropenia; (F) pulmonary hypertension; and (G) thrombocytopenia.

**Pulmonary hypertension.**   Three studies provided outcomes regarding pulmonary hypertension in patients with HH and were included in the meta-analysis. Significant evidence of heterogeneity among the studies was found ($P_{\text{heterogeneity}}$ = 0.013, $I^2$ = 76.8%, $P_{\text{effect}}$ = 0.005); therefore, a random-effects model of analysis was used. The pooled proportion of patients who had pulmonary hypertension was 2% (95% CI = 0–4) (Fig 3F).

**Thrombocytopenia.**   Two studies provided outcomes regarding thrombocytopenia in patients with HH and were included in the meta-analysis. Significant evidence of heterogeneity among the studies was found ($P_{\text{heterogeneity}}$ = 0.013, $I^2$ = 83.7%, $P_{\text{effect}}$ < 0.008); therefore, a random-effects model of analysis was used. The pooled proportion of patients who had thrombocytopenia was 2% (95% CI = –1 to 5) (Fig 3G).

## Publication bias

Funnel plot and Begg's and Egger's tests were performed to assess publication bias among the studies. The shapes of the funnel plots showed obvious evidence of asymmetry (Fig 4), and the $P$ value of Egger's test confirmed the existence of publication bias for the response to diazoxide (Begg's test $P$ = 0.462; Egger's test $P$ = 0.045). The trim-and-fill method showed no need for additional studies (Fig 5).

## Discussion

This systematic review and meta-analysis evaluated the efficacy and safety of diazoxide for patients with HH. Six cohort studies involving 1142 patients were included. This meta-analysis was novel in evaluating the efficacy and safety of diazoxide for patients with HH. The meta-analysis showed that diazoxide was potentially useful in HH management; however, it had

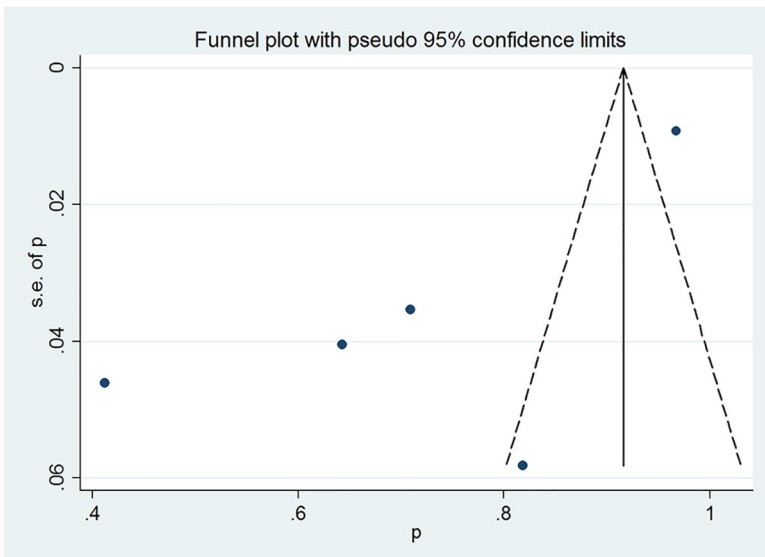

**Fig 4. Funnel plot of diazoxide responsiveness for testing publication bias.** Each point represents a separate study for the indicated association.

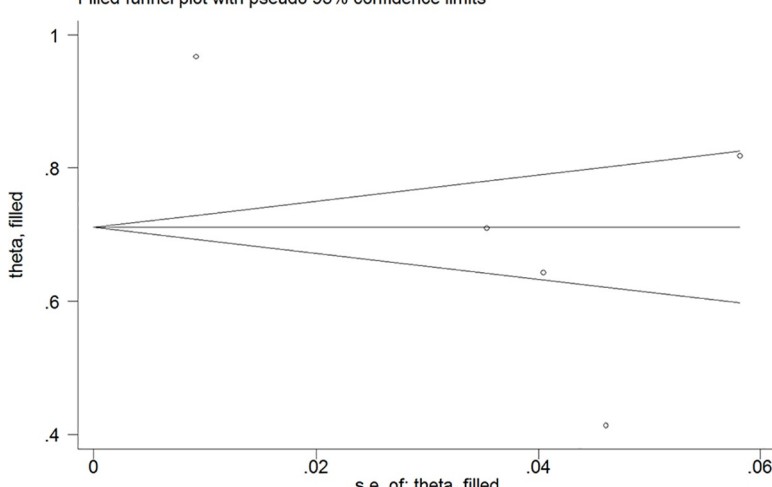

**Fig 5. Filled funnel plot of diazoxide responsiveness using the trim-and-fill method.**

some side effects needing careful monitoring. Better-designed randomized controlled trials are still required to confirm the findings.

Diazoxide is the first-line drug for hyperinsulinemia and the only drug approved by the Federal Drug Administration. Its use has increased over the years, including patients with various genetic forms of hyperinsulinemia or perinatal stress hyperinsulinemia as well as infants of mothers with diabetes who have received this treatment more than ever before [25]. In 1964, diazoxide was first reported as an effective treatment for leucine-sensitive hypoglycemia [26]. Subsequently, several studies reported its effectiveness in treating metastatic or inoperable insulinomas [27,28]. Diazoxide is usually effective in children with complete $K_{ATP}$ channels, but it is ineffective if the $K_{ATP}$ channels are malformed due to ABCC8/KCNJ11 mutation. However, the exception is that children with ABCC8 mutation respond to diazoxide, indicating that cellular adaptation and redundancy in the standard $K_{ATP}$ channel model determine the pathophysiology of CHI [29]. In this study, 71% (646/831) of patients responded to diazoxide treatment. Besides, about 25% of children with CHI were partially or completely unresponsive to diazoxide [30]. Other drugs, such as octreotide, may be required for the second-line treatment in such children [31].

The most common side effect is mild-to-severe hypertrichosis, which is thought to depend on the dose for each patient [18]. In this study, the most common adverse drug reaction was hypertrichosis (45%). For older children, body hair can be troublesome, and they may choose to use other drugs to avoid this complication. In theory, local $K_{ATP}$ channel blockers, such as toluene butylamine, may reduce hair growth, but this indication has not been systematically assessed [32]. The retention of sodium and fluorine is a common side effect of diazoxide (18%) [17]. All patients taking diazoxide should be carefully monitored for weight, electrolytes and edema, and cardiopulmonary function, especially when starting or increasing doses. The simultaneous use of diuretics, such as chlorothiazide, can reduce urinary retention. In 2015, the US Food and Drug Administration issued a drug safety statement because 11 infants treated with nitrous oxide developed pulmonary hypertension (PH) [33]. It is believed that about 2.4% of all children treated with diazoxide have PH [17,18,34]. These data were consistent with the results of the present study (2%). A recent study published by Herrera et al. [17] employed all patients with a formal diagnosis of hyperinsulinemia. Their study included perinatal stress–induced hyperinsulinemia and showed that 2.4% of patients developed PH after

starting the use of sodium diazoxide. They believed that PH is more likely to occur in preterm and low-birth-weight infants.

At the same time, some limitations of this meta-analysis should be emphasized. First, meta-analyses may be biased when literature searches fail to identify all relevant trials or subjectively apply selection criteria for including trials. To minimize these risks, thorough searches in multiple bibliographic databases were conducted, and clear criteria were used for research selection, data abstraction, and data analysis. Second, all the studies included in the present meta-analysis were observational. Observational studies are susceptible to selection bias and confusion, leading to the underestimation or overestimation of the actual effects of the intervention. Finally, some studies had small sample sizes, thus reducing the statistical power.

## Conclusions

In summary, the results demonstrated that diazoxide had a potential role in treating HH; however, it had some side effects needing careful monitoring. Furthermore, well-designed large-scale studies, such as randomized controlled trials, might be necessary in the future to obtain more evidence.

## Supporting information

**S1 Checklist. PRISMA 2009 checklist.**
(DOC)

**S1 Table. Search strategies for Embase.**
(DOC)

**S1 Data.**
(ZIP)

## Author Contributions

**Conceptualization:** Xiaohong Chen.

**Data curation:** Xiaohong Chen, Lifang Feng, Luhong Yang, Yuan Qin.

**Formal analysis:** Lifang Feng, Hui Yao, Yuan Qin.

**Investigation:** Lifang Feng, Hui Yao.

**Methodology:** Hui Yao.

**Project administration:** Xiaohong Chen.

**Resources:** Xiaohong Chen, Luhong Yang.

**Software:** Xiaohong Chen, Luhong Yang.

**Validation:** Yuan Qin.

**Writing – original draft:** Xiaohong Chen.

**Writing – review & editing:** Xiaohong Chen, Lifang Feng, Hui Yao, Luhong Yang, Yuan Qin.

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
