## [Decision Letter · Decision Letter 0]

2 Nov 2020

PONE-D-20-17226

Efficacy and safety of diazoxide for treating hyperinsulinemic hypoglycemia: a systematic review and meta-analysis

PLOS ONE

Dear Dr. Chen,

Thank you for submitting your manuscript to PLOS ONE. After careful consideration, we feel that it has merit but does not fully meet PLOS ONE’s publication criteria as it currently stands. Therefore, we invite you to submit a revised version of the manuscript that addresses the points raised during the review process.

We look forward to receiving your revised manuscript.

Kind regards,

Cho Naing, MBBS, PhD, FRCP

Academic Editor

PLOS ONE

Journal Requirements:

2. Please confirm that you have included all items recommended in the PRISMA checklist including the full electronic search strategy used to identify studies with all search terms and limits for at least one database.

3.In your Data Availability statement, you have not specified where the minimal data set underlying the results described in your manuscript can be found. PLOS defines a study's minimal data set as the underlying data used to reach the conclusions drawn in the manuscript and any additional data required to replicate the reported study findings in their entirety. All PLOS journals require that the minimal data set be made fully available. For more information about our data policy, please see http://journals.plos.org/plosone/s/data-availability.

Additional Editor Comments (if provided):

This is an interesting research area. The authors can improve the quality of their manuscript.

Introduction

The authors need to state the objective in the last paragraph, although it has been described in the Abstract

Methods

The authors need to improve this section.

Study selection

It will be informative to provide more details of search strategy used for Embase in appendix.

Study selections

The authors have indicated clinical studies as an inclusion criteria.

However, it has indicated in the abstract that cohort studies are included.

The quality assessment use was the NOS checklist. Hence, the included studies were observational type, rather than clinical studies. Please, provide what type of observational studies....... Only cohort or/ and case-control designs.

In Table 1, please add 1 more column for study design.

Data extraction

Please, indicate the initials of two authors who had done study selection/data extraction

Statistical analysis

It is essential to describe the outcome measurement ( e.g OR or RR and its 95%CI).

To be more informative, please define the response rate (e.g what is numerator/ denominator, etc)

Reference

In addition to #21 (Lu et al).......... It will be better to refer Higgins et al 2019 ( available from the Cochrane Library website)

.

Thank you

Reviewers' comments:

Reviewer's Responses to Questions

**Comments to the Author**

1. Is the manuscript technically sound, and do the data support the conclusions?

Reviewer #1: Yes

2. Has the statistical analysis been performed appropriately and rigorously? 

Reviewer #1: Yes

3. Have the authors made all data underlying the findings in their manuscript fully available?

Reviewer #1: Yes

4. Is the manuscript presented in an intelligible fashion and written in standard English?

Reviewer #1: Yes

5. Review Comments to the Author

Reviewer #1: Overall, the manuscript was well written, and presented adequate study methodology, results and discussion.

Some minor clarifications are required on the following.

Discussion

Para 2, Line 2: “Its use has increased… having more babies than ever before.” – This statement is unclear.

Para 2, Line 9: “However, the exception is that children with ABCC8 mutation respond to diazoxide, indicating cellular adaptation and red blood cells”- The statement regarding red blood is unclear

6. PLOS authors have the option to publish the peer review history of their article (what does this mean?). If published, this will include your full peer review and any attached files.

Reviewer #1: No

---

## [Author Response · Author response to Decision Letter 0]

9 Dec 2020

Dear editor,

On behalf of all co-authors, we would like to express our great appreciation to you and the reviewers for your positive, constructive and insightful comments on our manuscript entitled “Efficacy and safety of diazoxide for treating hyperinsulinemic hypoglycemia: a systematic review and meta-analysis”. These comments and suggestions are really helpful in improving the quality of our manuscript. Accordingly, we have revised the manuscript. In addition, point-by-point responses to the editor’s and reviewers’ comments are listed below.

We, again, thank you and the reviewers for your constructive comments and hope that the revision is acceptable for publication in your journal. 

We are looking forward to hearing from you soon.

Sincerely, 

Point-by-point responses to the editor’s comments:

Important Comments:

Comment 1: Introduction The authors need to state the objective in the last paragraph, although it has been described in the Abstract.

Response: We thank you for this kind reminding. Accordingly, we have added it to Introduction in the newly revised version.

Comment 2: Study selection It will be informative to provide more details of search strategy used for Embase in appendix.

Response: As suggested, we have added it in the newly revised version (Table S1).

Comment 3: The authors have indicated clinical studies as an inclusion criteria. However, it has indicated in the abstract that cohort studies are included. The quality assessment use was the NOS checklist. Hence, the included studies were observational type, rather than clinical studies. Please, provide what type of observational studies....... Only cohort or/ and case-control designs. In Table 1, please add 1 more column for study design.

Response: Thanks for this kind suggestion, we have added it to Table 1 in the newly revised version.

Comment 4: Data extraction Please, indicate the initials of two authors who had done study selection/data extraction.

Response: Thanks for this kind suggestion, we have added them in the newly revised version.

Comment 5: Statistical analysis It is essential to describe the outcome measurement ( e.g OR or RR and its 95%CI). To be more informative, please define the response rate (e.g what is numerator/ denominator, etc).

Response: Thanks for this kind suggestion, we have added them to Statistical analysis in the newly revised version.

Comment 6: Reference. In addition to #21 (Lu et al).......... It will be better to refer Higgins et al 2019 (available from the Cochrane Library website).

Response: Thanks for this kind suggestion, we have added it to Reference in the newly revised version.

Point-by-point responses to the reviewer’s comments:

Reviewer #1:

Important Comments:

Comment 1: Overall, the manuscript was well written, and presented adequate study methodology, results and discussion. Some minor clarifications are required on the following.

Response: It is pleasing to have acknowledged our diligence in completed this manuscript. We appreciate the reviewer’s kindly comments.

Comment 2: Discussion Para 2, Line 2: “Its use has increased… having more babies than ever before.” – This statement is unclear.

Response: Thanks for this kind suggestion, we have rewritten them in the newly revised version.

Comment 3: Para 2, Line 9: “However, the exception is that children with ABCC8 mutation respond to diazoxide, indicating cellular adaptation and red blood cells”- The statement regarding red blood is unclear.

Response: Thanks for this kind suggestion, we have rewritten them in the newly revised version.

---

## [Editor Report · Decision Letter 1]

14 Dec 2020

PONE-D-20-17226R1

Efficacy and safety of diazoxide for treating hyperinsulinemic hypoglycemia: a systematic review and meta-analysis

PLOS ONE

Dear Dr. Chen,

Thank you for submitting your manuscript to PLOS ONE. After careful consideration, we feel that it has merit but does not fully meet PLOS ONE’s publication criteria as it currently stands. Therefore, we invite you to submit a revised version of the manuscript that addresses the points raised during the review process.

We look forward to receiving your revised manuscript.

Kind regards,

Cho Naing, MBBS, PhD, FRCP

Academic Editor

PLOS ONE

Additional Editor Comments (if provided):

The current analysis was based on six cohort studies.

In the Background, six clinical studies. Please make a correction.

The inclusion criteria were as follows: (1) clinical studies focusing on ..................... This was incorrect. The authors have selected only cohort studies. No clinical studies. Plese update this. selection criteria. I have already highlighted this point in initial comments.

The authors still need substantial improvements in grammar/language throughout the text

Examples,

In abstrct

The meta-analysis of the efficacy and safety of diazoxidein treating HH was performed by searching the PubMed, Embase, ..............

Two missing words "relevant studies" between searching the and PubMed,

A total of 6studies, ................... It should be presented 'a total of six cohort studies'.,

Among the cohort studies, the pooled estimate of patients who were diazoxide responsive was 71% (95% CI 50%–93%.................. Please rephrase this sentence by adding pool estimates of what outcome.

In Text

A meta-analysis of cohorts based on the random-effects............................ It should describe 'cohort studies, instead of cohorts. .

---

## [Author Response · Author response to Decision Letter 1]

30 Dec 2020

Dear editor,

On behalf of all co-authors, we would like to express our great appreciation to you for your positive and constructive comments on our manuscript entitled “Efficacy and safety of diazoxide for treating hyperinsulinemic hypoglycemia: a systematic review and meta-analysis”. These comments and suggestions are really helpful in improving the quality of our manuscript. Accordingly, we have revised the manuscript. In addition, point-by-point responses to the editor’s comments are listed below.

We hope that the revision is acceptable for publication in your journal. 

We are looking forward to hearing from you soon.

Sincerely, 

Point-by-point responses to the editor’s comments:

Important Comments:

Comment 1: The current analysis was based on six cohort studies. In the Background, six clinical studies. Please make a correction.

Response: Accordingly, we have corrected it in the newly revised version.

Comment 2: The inclusion criteria were as follows: (1) clinical studies focusing on ..................... This was incorrect. The authors have selected only cohort studies. No clinical studies. Plese update this. selection criteria. I have already highlighted this point in initial comments.

Response: Accordingly, we have corrected it in the newly revised version.

Comment 3: The authors still need substantial improvements in grammar/language throughout the text.

Examples,

In abstract

The meta-analysis of the efficacy and safety of diazoxidein treating HH was performed by searching the PubMed, Embase, ..............

Two missing words "relevant studies" between searching the and PubMed,

A total of 6studies, ................... It should be presented 'a total of six cohort studies'.,

Among the cohort studies, the pooled estimate of patients who were diazoxide responsive was 71% (95% CI 50%–93%.................. Please rephrase this sentence by adding pool estimates of what outcome.

In Text

A meta-analysis of cohorts based on the random-effects............................ It should describe 'cohort studies, instead of cohorts. .

Response: Accordingly, we have carefully proofread the manuscript and corrected the spelling and grammatical errors in the revised manuscript.

---

## [Editor Report · Decision Letter 2]

6 Jan 2021

PONE-D-20-17226R2

Efficacy and safety of diazoxide for treating hyperinsulinemic hypoglycemia: a systematic review and meta-analysis

PLOS ONE

Dear Dr. Chen,

Thank you for submitting your manuscript to PLOS ONE. After careful consideration, we feel that it has merit but does not fully meet PLOS ONE’s publication criteria as it currently stands. Therefore, we invite you to submit a revised version of the manuscript that addresses the points raised during the review process.

We look forward to receiving your revised manuscript.

Kind regards,

Cho Naing, MBBS, PhD, FRCP

Academic Editor

PLOS ONE

Additional Editor Comments (if provided):

The authors have addressed almost all comments provided.

1) It is necessary to include your revised text in your reply to the comments.

2) The authors have to update literature search whether there may be additional published studies after your initial search in 2019. This is crucially important in this field as the evidence must be based on comprehensive review of all eligible studies.

Thank you
---

## [Author Response · Author response to Decision Letter 2]

18 Jan 2021

Dear editor,

On behalf of all co-authors, we would like to express our great appreciation to you for your positive and constructive comments on our manuscript titled “Efficacy and safety of diazoxide for treating hyperinsulinemic hypoglycemia: a systematic review and meta-analysis.” These comments and suggestions are really helpful in improving the quality of our manuscript. Accordingly, we have revised the manuscript. In addition, point-by-point responses to the editor’s comments are listed below.

We hope that the revision is acceptable for publication in your journal. 

Sincerely, 

Point-by-point responses to the editor’s comments:

Important Comments:

Comment 1: The current analysis was based on six cohort studies. In the Background, six clinical studies. Please make a correction.

Response: Thank you for the comments. We have corrected it in the revised version.

Revised text: “This meta-analysis based on six cohort studies was conducted and aimed to investigate the efficacy and safety of diazoxide for treating CHI.”

Comment 2: The inclusion criteria were as follows: (1) clinical studies focusing on ..................... This was incorrect. The authors have selected only cohort studies. No clinical studies. Please update this. selection criteria. I have already highlighted this point in initial comments.

Response: Thank you for the comments. We have corrected it in the revised version.

Revised text: “Cohort studies focusing on the efficacy and safety of diazoxide in treating HH.”

Comment 3: The authors still need substantial improvements in grammar/language throughout the text.

Examples,

In abstract

The meta-analysis of the efficacy and safety of diazoxidein treating HH was performed by searching the PubMed, Embase, ..............

Two missing words "relevant studies" between searching the and PubMed,

A total of 6studies, ................... It should be presented 'a total of six cohort studies'.,

Among the cohort studies, the pooled estimate of patients who were diazoxide responsive was 71% (95% CI 50%–93%.................. Please rephrase this sentence by adding pool estimates of what outcome.

In Text

A meta-analysis of cohorts based on the random-effects............................ It should describe 'cohort studies, instead of cohorts. .

Response: Thank you for the suggestion. We have carefully proofread the manuscript and corrected the spelling and grammatical errors in the revised manuscript.

Revised text: “The meta-analysis of the efficacy and safety of diazoxide in treating HH was performed by searching relevant studies in the PubMed, Embase, and Cochrane databases.”

“A total of 6 cohort studies, involving 1142 participants, met the inclusion criteria.”

“Among the cohort studies, the pooled estimate of the response rate of diazoxide therapy was 71%.”

“A meta-analysis of cohort studies based on the random-effects model was conducted to evaluate the clinical response rate.”

Comment 4: The authors have to update literature search whether there may be additional published studies after your initial search in 2019. This is crucially important in this field as the evidence must be based on comprehensive review of all eligible studies.

Response: Thank you for the suggestion. The literature search has been updated up to January 2021.

Revised text: “The PubMed, Embase, and Cochrane databases were searched for studies published up to January 2021.”

“The electronic search identified 348 studies. Three additional studies were found by hand searching from the reference lists of other review studies. According to the inclusion criteria, 227 studies remained after removing the duplicates. Subsequently, 193 irrelevant studies were excluded. Of the remaining 34 studies, 15 were letters, reviews, and meta-analyses, and hence excluded. The remaining studies were systematically reviewed, and 19 were qualified for full-text reading. After full-text reading, five studies not focusing on children, three lacking usable data, and five case reports were excluded.” (Figure 1)

---

## [Editor Report · Decision Letter 3]

20 Jan 2021

Efficacy and safety of diazoxide for treating hyperinsulinemic hypoglycemia: a systematic review and meta-analysis

PONE-D-20-17226R3

Dear Dr. Chen,

We’re pleased to inform you that your manuscript has been judged scientifically suitable for publication and will be formally accepted for publication once it meets all outstanding technical requirements.

Kind regards,

Cho Naing, MBBS, PhD, FRCP

Academic Editor

PLOS ONE

Additional Editor Comments (optional):

The authors have addressed the comments. Thank you
---

## [Editor Report · Acceptance letter]

2 Feb 2021

PONE-D-20-17226R3 

Efficacy and safety of diazoxide for treating hyperinsulinemic hypoglycemia: a systematic review and meta-analysis 

Dear Dr. Chen:

I'm pleased to inform you that your manuscript has been deemed suitable for publication in PLOS ONE. Congratulations! Your manuscript is now with our production department. 

Kind regards, 

on behalf of

Professor Cho Naing 

Academic Editor

PLOS ONE